# The Potentials for the Ecological Management of Landscape Connectivity Including Aquatic Ecosystems in Northeast Albania

**Laura Shumka** [1,*]**, Andi Papastefani** [1]**, Spase Shumka** [2] **and Sotir Mali** [3]

1 Department of Art and Design, Albanian University, Tirana 1001, Albania
2 Department of Biotechnology and Food, Agricultural University of Tirana, Tirana 1025, Albania
3 Department of Biology, University of Elbasan, Elbasan 3000, Albania
* Correspondence: shumkalaura@gmail.com

**Abstract:** At the landscape level, freshwater ecosystems are linked at various spatial and temporal scales by movements of different fauna components adapted to life in water. We review the literature on the state of landscape connectivity and related aquatic species that connect different types of freshwater habitats, focusing on linkages from streams, large rivers and standing water bodies. Based on existing evidence, it is clear that biotic linkages throughout current mosaic have important consequences for biological integrity and biodiversity. The recent developments with regard to urbanization, expansion of urban centers, infrastructure development, and hydropower plant construction in Albania are in line with global destruction and fragmentation of habitats resulting in the parceling up of landscapes that, in this very case, have been caused by human population growth and development activities. The primary aim of this article is to address the landscape connectivity in a wider northeastern part of Albania considering various protected areas. The landscape connectivity is a pillar component of connectivity conservation that has emerged as a response approach to a range of threats to biodiversity, which include habitat degradation and destruction, fragmentation and climate changes. The approach analyses of landscape connectivity were defined from a human perspective in a linkage among different protected areas, including National Park Albanian Alps, Nature Park Korrab-Koritnik, National Park ShebenikJabllanica, Pogradec Landscape Protected Areas and National Park Prespa. The basis of this analysis lies in the Network of Protected Areas (NPAs) of Albania. Cumulatively, the protected areas connectivity, aquatic ecosystem linkage and individual movements connect populations within and among landscape mosaics and contribute to national and regional diversity and resilience to disturbance. This study highlights the importance of considering both terrestrial and aquatic ecosystems connectivity in conservation planning and management.

**Keywords:** biotic integrity; river/streams; aquatic ecology; wetlands; biota

## 1. Introduction

Despite the developments, landscapes that retain more connections between patches of otherwise isolated areas of vegetation [1], which therefore have higher levels of landscape connectivity, are assumed to be more likely to maintain populations of various species that inhabited the original landscape [1,2]. Further, the lack of landscape connectivity can have a range of negative impacts on ecosystem functioning. It may result in vegetation patches remaining unoccupied for suites of species [1,3], meaning that the spatial distribution of these taxa may not directly correspond to the spatial distribution of available habitat for them [3,4]. In the case of our survey focus area, this is illustrated by some terrestrial and aquatic species, as Balkan lynx (*Lynx lynx balcanicus*) facing serious reduction and isolation [5] or freshwater fish species such as Skadar gudgeon (*Gobio scadarensis*) and Ohrid spirlin (*Alburnoides ohridanus*) struggling from the conversion of the aquatic ecosystem for running to standing one.

Recent analysis findings [6] reveal that the Government of Albania has approved a System of Environmentally Protected Areas. Currently, the area of the Network of Protected

Areas (NPA) of Albania has reached 504,826.3 ha, or 21% of the total area of the country. Of the total area, the Coastal and Marine Protected Areas constitute 119,224.7 ha, or 23.6% of the total surface of the NPAs of the country, of which 13,261.2 ha is only marine area. Also, 98,180.6 ha, are with the status of Ramsar areas, which cover 3.42% of the total area of the country. The effective conservation management of protected areas is a prerequisite for their connectivity performance [7–9], while secured conservation connectivity provides opportunities for species survival and performing of life cycles. Further on the connectivity of protected areas systems, it is necessary to facilitate large-scale ecological and evolutionary processes, such as gene flow, migration and species range shifts. These processes are all essential for the persistence of viable populations, especially when facing climatic and environmental changes in increasingly transformed and fragmented landscapes [10]. Improving or sustaining protected area connectivity is therefore a primary concern for the effective conservation and management of biodiversity [11].

On the map of the Mediterranean region, the Balkan Peninsula is one of the sub-regions representing one of the world's 25 biodiversity hotspots [12,13] (Myers 2000; Weis, et al., 2018). In the map of global ecoregions [14] (Abell et al., 2008), the Black Drin River and all other tributaries located in the survey focus area are part of the 420-Southeast Adriatic Drainage. It is known that this map of freshwater ecoregions is based on the distributions and compositions of freshwater fish species and incorporates major ecological and evolutionary patterns. Besides this, the complexity of Drini basin and adjacent water drainage makes the delineation a challenging issue. Discharge of all Albanian rivers is seasonally highly variable, being sometimes more than ten times less in summer than in winter. The beds of the main rivers are usually very wide, as a great amount of gravel and pebbles is deposited around the flow itself [15].

The National Agency of the Protected Areas (NAPA) and the structures at the local level have a great responsibility and challenge to face the current situation and the perspective related to protected areas and their management. This is also due to the fact that protected areas in Albania are evidenced in various shapes and sizes (land, water, sea, local and cross-border); in public, municipal and private ownership; in six categories of administration; Ramsar wetland area of international importance; Biosphere Reserves and as UNESCO World Heritage Sites, i.e., a complexity and natural heritage that should be clearly reflected in the NAPA program.

The establishment of the Albanian Ecological Network, i.e., NPA, is based on the fact that networks of connected areas have formed the basis for establishing corridors that extend across regional areas to even wider country contexts and trans-boundary ones (Figures 1 and 2). In Figure 2, the protected areas along the potential landscape corridors are presented.

There are two other aspects related to the connectivity of the PAs at the project focus area in Albania and three selected corridors. Firstly, the connection lies at the eastern peleogeographic domains of Albanides [16]. Secondly, it is connected to the Dinaric regions. Definitions of the Dinaric Region differ between sources. In its narrowest sense, the region can be restricted to Dinaric Karst. Following recent approaches [17], taking into consideration the fact that the northern Albanian Alps are geographically a part of the Dinaric region as defined by the WWF, the scope of the area and its borders are unavoidably arbitrary. Thus, they are extended and the south-eastern border is also artificial since it follows a political division between Albania and North Macedonia.

Even fragmented, the analysis of threats conducted by using different methodologies, such as the Management Effectiveness Tracking Tool [18]; World Heritage Outlook assessment [19] or Bird Life International's Important Bird and Biodiversity Area (IBA) monitoring protocol [20], have identified a range of threats affecting the integrity of the protected areas.

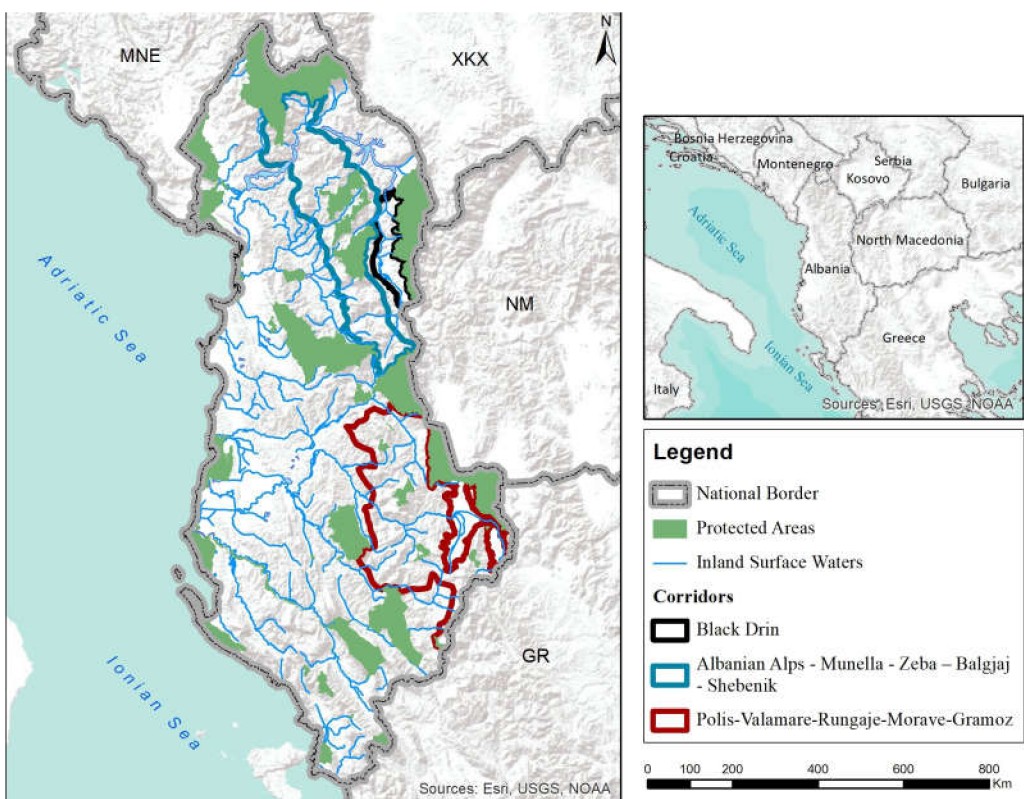

**Figure 1.** Map of Albania considering the potential landscape corridors (Melovski et al., 2022).

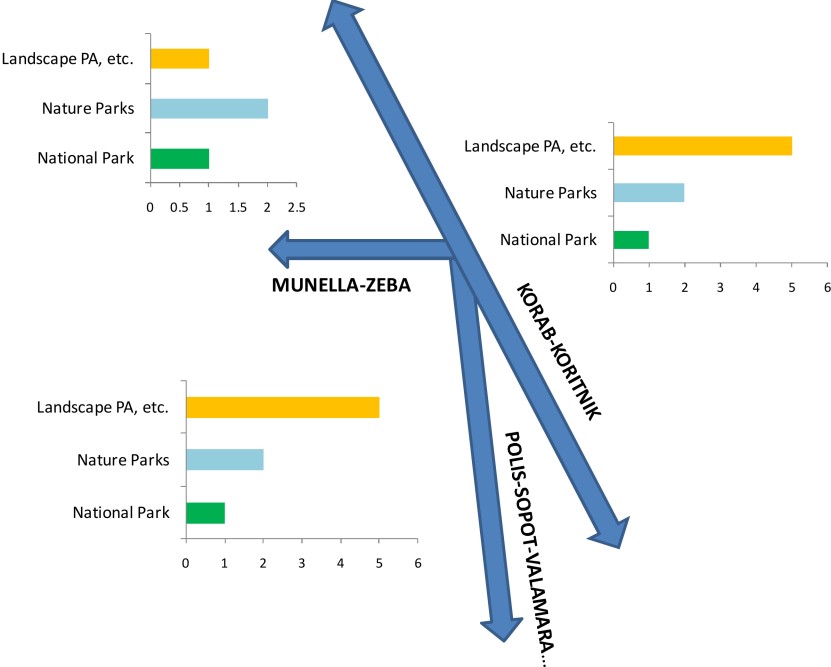

**Figure 2.** Protected areas along the selected corridors.

## 2. Materials and Methods

For conducting this survey, we searched for relevant references (books, peer-reviewed articles, white papers, linkage designs, theses and reports) using Cambridge Scientific Abstracts, ISI Web of Science, and Scopus, searching various combinations of keywords related to landscape connectivity, aquatic ecosystems, roads, canals, hydropower plants, rivers, streams, riparian areas, urbanization, artificial water bodies, i.e., reservoirs, high-

ways, livestock grazing, and agriculture in combination with keywords related to habitats, biodiversity, conservation, linkages, and corridors.

Following the above mentioned analyses, we found 2540 documents relevant to the survey topic, 67 documents directly linked to the ecological landscape conservation in the Balkans and considered 40 for conducting the article.

For the GIS approach and use of the protected areas layer for connecting potential corridors, current land use and intervention was performed as well. Different IUCN categories of protected areas were analyzed, while the area covered by them towards the entire area was calculated as well.

The map was built in the Arc Map 10.3 program with the WGS 84 coordinate system. The relief map of Albania (1:20,000), the Albanian hydrographic network (obtained from asig.gov.al (accessed on 29 December 2022)), was used as a base map; map of the country protected areas.

Further, we used reference sources to recommend management practices at the Section 5 towards conservation needs and avoidance of landscape fragmentation barriers, streams and riparian areas, and farms to conserve connectivity and ecosystem processes on a landscape scale.

### 3. Results

The data presented in Figures 1 and 2 show that the actual protected areas system represents a good starting point for building the landscape connectivity at three major axes, where they represent, in the Korab-Koritnik-Albanian Alps, 45% of the entire area (including non protected one), and at the area of Munella-Zeba, 40% and 30% of the entire area of the selected Polis-Sopot-Valamara. An important component within the entire system represents Drini River with its associated tributaries. Based on ground analyses and landscape analyses, the following are major threats that affect habitat and species connectivity at the north-eastern part of Albania: (i) Dam construction, energy and mining projects; (ii) transportation and service corridors; (iii) residential and commercial development; (iv) tourism; (v) natural systems modification; (vi) biological resources use; (vii) alien and invasive species; (viii) pollution; (ix) climate change and severe weather; (x) agriculture and aquaculture; (xi) deforestation; and (x) forest fires, etc. The energy infrastructure such as constructions of dams and mining are among the frequently present threats to PAs and with a high impact, compared to other threats. For the most frequent level of one threat, i.e., biological resource use threats, natural system modifications, etc., seems to have a high impact.

In Figure 2, three selected corridors intersecting important freshwater ecosystems are presented where there is protected areas coverage (different categories including three National Park, six Landscape Protected Areas and eleven Nature Parks). Physical barriers between different aquatic habitats, such as reservoirs, mountain ranges, dams, or intervening inhospitable habitats, are restricting movements required to establish or maintain biological connectivity for both terrestrial and aquatic ecosystems. In the case of freshwater habitats and in a situation when all other factors, such as climate, topography, and geology are equal, large, high-quality aquatic habitats separated frequently (mostly within Corridor Korab-Koritnik, the northwest-southeast direction) by dams and other infrastructure elements are more likely to be a challenge for biologically connectivity, due to limited carrying capacity of not affected components (Figure 2).

### 4. Discussion

#### 4.1. Corridors Biota Connect Aquatic and Terrestrial Habitats throughout Freshwater Ecosystem

Greater spatial distance between suitable habitats may increase the number and variety of intervening landscape patches through which organisms must move. This has also been presented [21] as an element of decreasing the probability of traversing them successfully [22]. Further on, biological connections depend on the biota present in the system. The physical structure of the landscape mosaic ecosystems, including freshwater,

is one that determines the system's structural connectivity; the species present determine how structural connectivity is translated into actual or functional connectivity [23]. Species traits and individual behaviors, such as dispersal mode, dispersal propensity, life cycle requirements, and responses to disturbance or environmental cues, arise over time in response to abiotic and biotic selection pressures.

As defended by [24], the freshwater ecosystem mosaics in the case of three selected conservation corridors include rivers, streams together with lakes, ponds, and other freshwater habitats, which are the diverse collection of integrated freshwater habitats needed to sustain aquatic life and the ecological integrity of these systems [25]. The most significant mosaic is that of the Drini watershed that connects, at the transboundary scale, an incredible number of aquatic bodies. A long time ago, the natural freshwater connectivity Adriatic Sea-Lake Ohrid and adjacent wider area has been affected. The fragmentations/dam and impoundments establishment created massive reservoirs that flooded large parts of the Drini valley and altered the natural habitats. The Albanian Alps region was partly disconnected from the mountain ranges further south due to the creation of three massive wide and deep lakes that were not present before the dams.

Thus, the connectivity within three selected corridors is secured via River Drini and it has a large network of streams (Korrab-Koritnik); streams/tributaraies of Rivers Drini and Mati (Munella-Zeba) and streams/tributaraies of Rivers Shkumbini, Osumi, Devolli, Vjosa (Polis-Sopot-Valamara-Gramoz).

### 4.2. Aquatic Connectivity in Landscape Settings

Recently, knowledge and understanding of the diversity and distribution patterns of freshwater fishes in most of the European Mediterranean has increased considerably. Nevertheless, the diversity, distribution, and conservation status of freshwater fish in some areas are still very poorly known, with the least known being in Albania. However, for the surrounding areas, updated information exists [26] but such data on Albanian species are scarce, apart from data on loaches (Cobitidae and Nemacheilidae) [27], salmonids [28] and barbels (genus *Barbus*; Cyprinidae) [29]. The available sources of information are the general works included in [30], which included 36 freshwater species and [31] included 77 species. The difference between the coverage in these two publications is probably due, in part, to the inclusion of newly introduced species, but more so by changes in the taxonomic status of many species. Moreover, both [32,33] include many doubtful taxa. The deficiency in the knowledge of the diversity of freshwater fishes of Albania has been confirmed by recent descriptions of many new species from the area [34,35].

The native fish populations and particularly endemic fish species in the Drini river system (Table 1-all littoral countries) are threatened by several anthropogenic activities and factors like:

- Water pollution caused mostly due to a lack of the wastewater treatment facilities as well as a lack of integrated management approaches;
- Relatively unregulated fishery practices and illegal fishing, use of destructive methods of fishing;
- Non-native fish species, accelerated abundance with unpredicted sequences to native endemic species; impacts on specific spawning grounds for specific species particularly due to serious impacts caused by water use in the agriculture sector with a constant presence of run-offs and no abatement plans;
- Poor integration of fishery management practices into the entire management of the area (including protected one as Nature Park Korrab-Koritnik, etc.), which is recognized internationally for its rich biodiversity and abundance of species, proclaimed as an important area for the conservation of European species and habitat., and IBA;
- Low rate of local awareness for the fish biodiversity, conservation threats. The awareness and knowledge are limited to a couple of commercial fish species.

**Table 1.** Fish species recorded during the survey and current conservation status.

| Species | English Name | IUCN Global | Albanian Red List | Habitat Directive | Bern Convention | Albanian/Balkan Endemics |
|---|---|---|---|---|---|---|
| *Alburnoidesohridanus* (Karaman, 1928) | Ohridspirlin | VU (D2) | | | | Ohrid Lake/Balkan endemic |
| *Alburnusscoranza* (Heckel et Kner, 1858) | Scadar Bleak | LC | | | | Balkan endemic |
| *Barbatulasturanyi* (Steindachner, 1892) | Brook loach | LC | | | | Ohrid Lake/Balkan endemic |
| *Barbus rebeli* (Köller, 1925) | Ohrid barbell | LC | LR | | | Balkan edemic |
| *Carassius gibelio* (Bloch, 1782) | Prussian carp | NE | | | | |
| *Chondrostomaohridanus* (Karaman, 1924) | Ohridnasse | DD | | | | Balkan endemic |
| *Cobitisohridana* (Karaman, 1928) | Ohrid spined loach | LC | LR | | | Balkan endemic |
| *Cypriniscarpio* (Linnaeus, 1758) | Carp | VU (A2ce) | | | | |
| *Eudentomyzonstankokaramani* (Karaman, 1974) | Drini brook lamprej | LC | | II | | Drini River |
| *Gobio scadarensis* (Karaman, 1924) | Scadar gudgeon | VU (D2) | LR/VU (D2) | | | Balkan endemic |
| *Hypophthalmichthys molitrix* (Valenciennes 1844) | Silver carp | Alien | | | | |
| *Oncorhynchus mykiss* (Walbaum, 1792) | Rainbow trout | Alien | | | | |
| *Pachychilonpictum* (Heckel et Kner, 1858) | Ohrid roach | LC | | | III | Balkan endemic |
| *Pelasgus minutes* (Karaman, 1924) | Ohridminow | DD | | | | Ohrid Lake/Black Drini |
| *Percafluviatilis* (Linnaeus, 1758) | European perch | LC | | | | |
| *Phoxinus Lum/ohridanus?* | Italian Minow spp | LC | | | | |
| *Pseudorasbora parva* (Temmini&Schlegel, 1846) | Stone moroke | Alien | | | | |
| *Rhodeusamarus* (Bloch, 1782) | Bitterling | LC | | II | III | |
| *Rutilus ohridanus* (Karaman, 1924) | Roach | DD | | | | Balkan endemic |
| *Salariafluviatilis* (Asso, 1801) | Freshwater blenny | LC | | | III | |
| *Salmo farioides* (Karaman, 1937) | Brown trout | LC | VU | | | |
| *Squaliusplatyceps* (Bonaparte, 1837) | Chub | LC | | | | Balkan endemic |
| *Sander lucioperca* (Linnaeus, 1758) | Pike-perch | LC | | | | |

Today, it is well recognized that the large rivers and their riparian zones are hot spots of biodiversity (Figure 3). According to [36], the fluvial geomorphic processes provide the habitat diversity and the specific habitat conditions for characteristic species assemblages and result in high levels of habitat diversity, local species richness (α-diversity), between-habitat differences (beta-diversity), and consequently, the overall species richness of a river.

It is widely accepted that freshwater fish species are seriously threatened by various factors [37]. Black Drini River is located at very interesting geographical settings within the wider Drini basin and has one of the most diverse ichthyophaunas comparable to other large rivers and lakes at the Balkans and northern Greece. It seems that fish are the most diverse group among all vertebrate groups in the river basin. Following the extremely

serious threats in the last three-to-four decades, most of the local native fish species have been impacted by human pressure, while some serious threats are still undergoing at the present day (Figures 4 and 5).

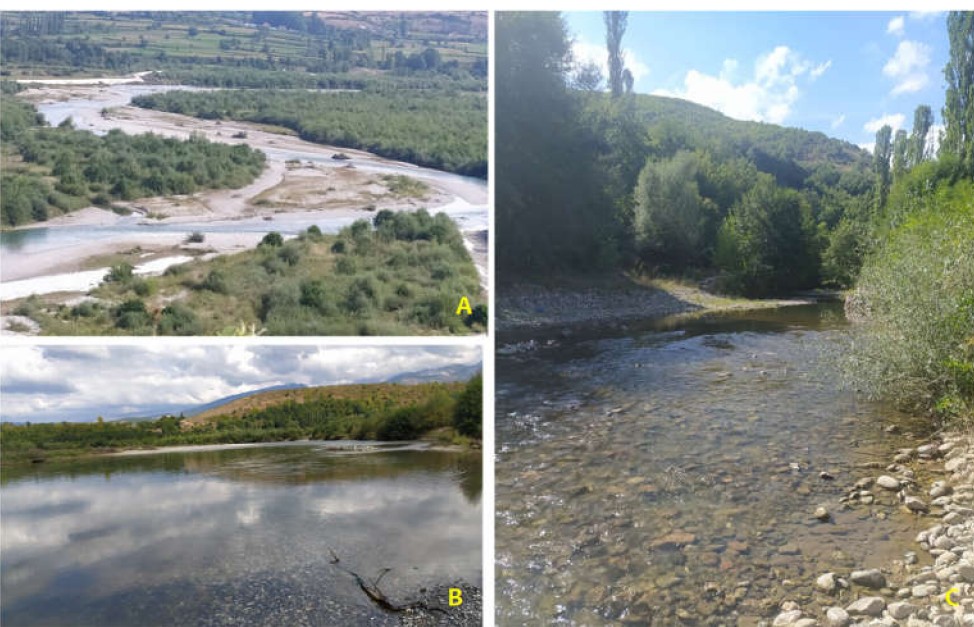

**Figure 3.** Fish sampling localities. (**A**) Fushe Cidhem; (**B**) Zall Rec and (**C**) Ujmisht.

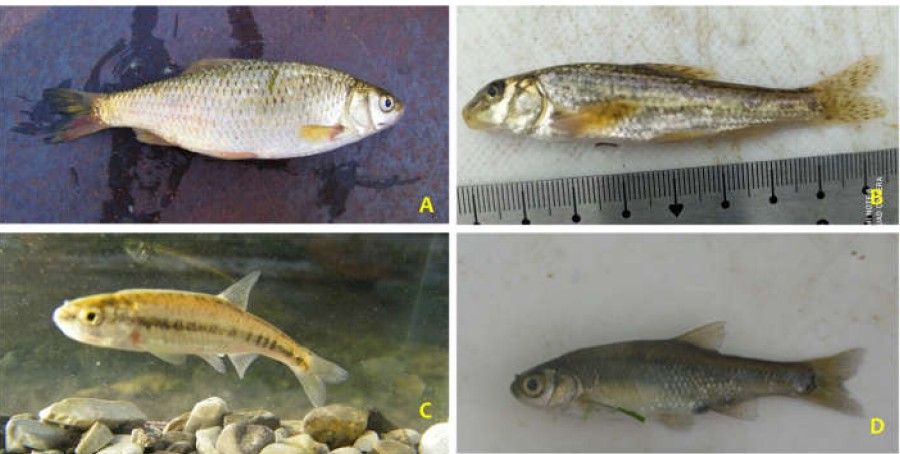

**Figure 4.** (**A**) *Rutilus ohridanus*; (**B**) *Gobio scadarensis*; (**C**) *Phoxinus* sp. *ohridanus*; (**D**) *Pelasgus minutes*.

The current conservation status of Drini fishes is presented in the following [38].The red listing process for Albanian fishes (Table 2) is poorly assessed, while in the North Macedonian part, the relevant Red data book is still under preparation.

The fish assemblage of Black Drini River is rapidly changing, and similar to the wider Mediterranean area, it is expected that it will follow in a situation of increased anthropogenic impacts and climate changes by the introduction of alien species. Being situated inproximity toconnected Lakes, neighboring ones and associated tributaries and systems where the assemblages are rapidly changing, there is a relatively high risk of changes regarding the composition and share. In our case with alien species, we must consider both exotic ones and those translocated from other ecoregions.

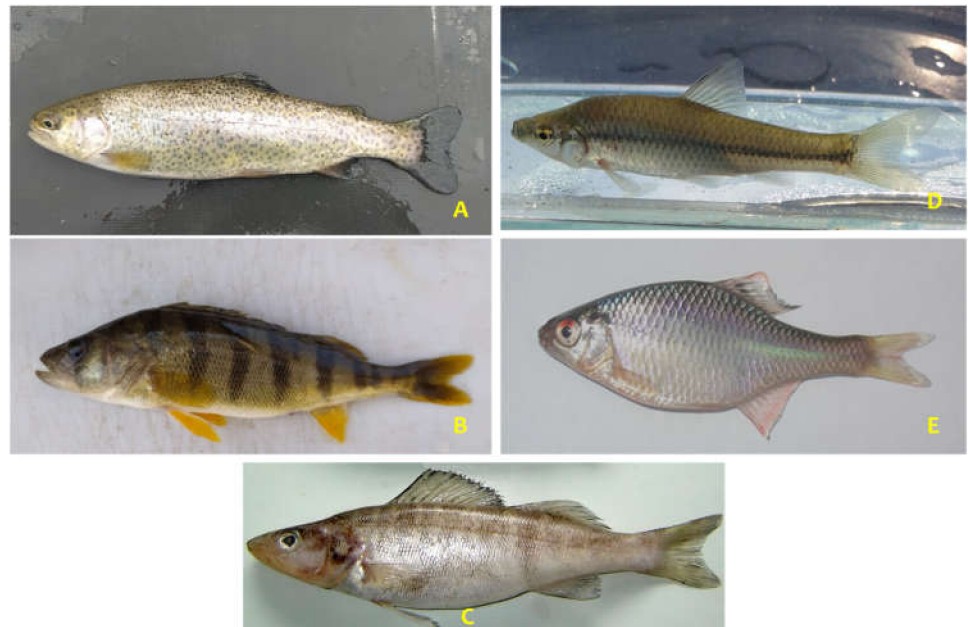

**Figure 5.** Alien species recorded in the River Black Drini survey locations. (**A**) *Onchorhyncis mykiss*; (**B**) *Perca fluviatilis*; (**C**) *Sander luciperca*; (**D**) *Pseudorasbora parva* and (**E**) *Rhodeus amarus*.

**Table 2.** Conservation status of fish in the Black Drini River.

| Threat Status | IUCN Red List | Albanian Red List |
|---|---|---|
| Critically Endangered (CR) | 1 | - |
| Endangered (EN) | - | - |
| Vulnerable (VU) | 4 | 2 |
| Near Threatened (NT) | - | - |
| Least Concern (LC) | 12 | 2 |
| Data Deficient (DD) | 3 | - |
| Not Evaluated (NE) | 3 | 19 |

*4.3. Legal Aspects of Fish and Ecosystem in Black Drini River Catchment*

The legislative framework for the fisheries and aquaculture sector includes several laws and by-laws. The main law regulating this sector is Law no 64/12 of 2012 "On Fishery". Albania is in the process of becoming an EU Candidate Country, and in this regard, is also in the process of aligning its legislation with the EU communitarian laws. Several by-laws have been approved that transpose some of the principles of the Common Fishery Policy into Albanian legislation. The legislation also contains the main principles of the Code of Conduct for Responsible Fisheries and has initiated the formation of the Fishery Management Organization for marine and inland waters. The law on fisheries has been further completed by laws in full accordance with EU Regulations and GFCM recommendations. The Albanian fisheries strategy from 2016 emphasizes good governance, sustainability, a competitive fleet where capacity is in balance with the resource, and alignment with the relevant EU legislation.

The legislation framework in the fishery sector, in an overall overview, is complete and advanced. The legislation deals not only with fishery issues but also with other related issues such a sbiodiversity, socio-economic aspects, etc.

In terms of conservation measures and designations, the River Black Drini catchment on theAlbanian side includes several protected areas, such as: Shebenik-Jabllanica National Park (II IUCN), (small part including tributaries), Lure-Mali i Dejes National Park (II),

Nature Managed Reserve Korab-Korritnik (II IUCN), Nature Managed Reserve Tej Drini i Bardhë(III IUCN), Managed Resources Protected Area of Luzni-Bulaç (III IUCN) and numerous Nature Monuments (III).

*Carassius gibelio*(Prussian carp) appeared in the Drini system (section of our survey) in the mid 1970′s [39,40], and following communication with local fisherman, it was very rare until 1980. The population expansion thereafter was followed by an invasion in the Fierza itself and also the river system, while it is rarely found in its tributaries. The current catch statistics place Prussian carp (beside that the data are not reliable) after bleak, common carp and common perch. During our survey in Keneta eKashtes, we recorded 12 individuals (range 8–14 mm) with an entire weight of 212 gr/100 m$^2$. The invasive potential in similar situations is explained by its specific reproductive flexibility (gynogenesis) [26].

There is an increase in the presence of Stone moroko (*Pseudorasbora parva*) all over the country, while our data from September to October 2021 revealed a low share in all localities with sings of increase at the localities close to the standing system, i.e., Fierza Lake.

It is not clear when the Pikeperch (*Sander lucioperca*), the most predatory fish in the study area, was introduced. There are strong believers that Zander was entering via White Drini, while following communication with local fisherman's, its population is oscillating. It appears that its prey food (mostly bleak, *Alburnusscoranza*) is quite stable and other reasons should be strived for explaining the Zanders patterns. During the survey, we recorded its presence in localities inSurroj and Këneta e Kashtës.

The ecological corridors cannot be understood without cultural contexts of the area that has specific value and advantages [24,25].

## 5. Conclusions

Given the fact that all ecological corridors (both terrestrial and aquatic) are shelters of important habitats and species, for areas in, or adjacent to, wildlife connectivity on landscape prospective, our recommendations are to: (1) Minimize the large scale intervention projects such as hydropower plants, roads and mining activities; (2) minimize the number and intensity of human activities within the area such as forest use, mining, etc, (3) maintain or re-establish natural processes in linkages, (4) establish sustainable development zones within lands and human-use areas, and (5) initiate road over cross for helping wildlife to avoid current barriers, and (6) encourage local government to unite and protect areas administration to incorporate the landscape connectivity into planning as a tool for integrating different practices in conservation approaches.

Landscape connectivity and conservation is a current approach to managing diverse ecosystems. In today's rapidly changing Balkan area, it is very challenging for protected areas on their own to properly conserve biodiversity. There is an urgent need to understand and effectively manage protected areas as part of the surrounding, and adapt to climate changes and developments.

**Author Contributions:** Conceptualization L.S. and S.S.; methodology, L.S. and S.S.; validation, L.S., S.S., A.P. and S.M.; formal analysis, L.S.; investigation, L.S. and S.S.; resources, L.S.; data curation, L.S.; writing-original draft preparation, L.S.; writing-review and editing, L.S.; visualization, L.S. and S.S.; Supervision, S.S.; project administration, L.S.; funding acquisition, L.S. All authors have read and agreed to the published version of the manuscript.

**Funding:** This research received no external funding.

**Institutional Review Board Statement:** Not applicable.

**Informed Consent Statement:** Not applicable.

**Data Availability Statement:** Readers can contact authors for availability of data and materials.

**Conflicts of Interest:** The authors declare no conflict of interest.

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
