# Peer review of "The Potentials for the Ecological Management of Landscape Connectivity Including Aquatic Ecosystems in Northeast Albania"

_2673-9917, doi:10.3390/hydrobiology2010004_

Round 1

Reviewer 1 Report

The manuscript was revised "The Potentials for Ecological Management of Landscape Connectivity Including Aquatic Ecosystems in Northeast Albania" prior to proceeding authors should consider the following comments:

1. Do not repeat the words of the title in the keywords.

2. Describe punctually and clearly in the abstract the tools used in the methodology.

3. The figure needs to increase the scale, the north and a wider reference on the world map (the lower map should be at a larger scale).

4. The authors state that the search was performed in WoS and Google Scholar, I think this overlaps, the results of Google Scholar will surely include WoS documents as well. I recommend performing the review in WoS and Scopus and in some regional database considering the study area.

5. The results are very brief, the authors should diagram and above all explain at least in about three paragraphs considering the objectives of their study.

Author Response

Response to reviewers

Reviewer 1

  1. Do not repeat the words of the title in the keywords.

The suggestion has been followed and correction done in the revised manuscript (page 1-Abstrac section of revised manuscript)

  1. Describe punctually and clearly in the abstract the tools used in the methodology.

The comment and suggestion was followed with abstract revision (page 1-Abstract)

  1. The figure needs to increase the scale, the north and a wider reference on the world map (the lower map should be at a larger scale).

The Figure 1 was completely revised following reviewer suggestion and comment (page 3 of revised manuscript)

  1. The authors state that the search was performed in WoS and Google Scholar, I think this overlaps, the results of Google Scholar will surely include WoS documents as well. I recommend performing the review in WoS and Scopus and in some regional database considering the study area.

This comment has been followed, and regional database considering the study area has been inserted (pages 3, for the regional contexts the expanded explanation at the introductory section)

  1. The results are very brief, the authors should diagram and above all explain at least in about three paragraphs considering the objectives of their study.

Results section expanded in pages 4 and 5 within current revised manuscript.

Reviewer 2 Report

The abstract must be redrafted. In this form, it does not contain any information about the test results. The introduction should contain background information about the topic. Currently, the introduction begins with a description of the research area, which is incomplete. The map, which is figure 1, has no scale, no legend, no description. The research area should be characterized in terms of physicogeography and hydrographics. Figure 2 requires more discussion. There is no description of the purpose of the research in the introduction. The description of the research area, methodology and results is unacceptable. Everything should be expanded and described in detail. The summary section is prepared correctly. The list of references is complete.

Author Response

Reviewer 2

  1. The abstract must be redrafted. In this form, it does not contain any information about the test results.

The comment was well understood and the abstract was redrafted (page 1-Abstrac section of revised manuscript)

  1. The introduction should contain background information about the topic. Currently, the introduction begins with a description of the research area, which is incomplete.

The suggestion has been followed with further background information incorporated (Pages 1, 2 and 3)

  1. The map, which is figure 1, has no scale, no legend, no description.

The map has been redrafted and adjusted following reviewer comments (Page 3)

  1. The research area should be characterized in terms of physic-geography and hydrographics.

Further data were inserted in-line to regions particularities (page 1,2 and 3)

  1. Figure 2 requires more discussion.

The suggestion was followed with further discussion outlined in pages 4 and 5

  1. There is no description of the purpose of the research in the introduction. The description of the research area, methodology and results is unacceptable.

Following this comment the sections were expanded, corrected and re-drafted.

Round 2

Reviewer 1 Report

I believe that the authors should be clear in their methodology. It is necessary to clearly describe in a table the search parameters and the total number of documents found and analyzed.

The map should highlight the country under study.

Author Response

Response to Reviewer 1

The comments and suggestions of the reviewer are the followings:

 I believe that the authors should be clear in their methodology. It is necessary to clearly describe in a table the search parameters and the total number of documents found and analyzed.

The comment of the reviewer has been well understood, so we reflected within current version of manuscript (page 4) the searched relevant references via Cambridge Scientific Abstracts, ISI Web of Science and Scopus.

Further on the method section was extended with map design approach.

In the previous revised version the English texts has been corrected by language expert.

The conclusions were expanded reflecting the survey approach and way forward. 

Reviewer 2 Report

Authors have corrected the manuscript in accordance with all suggested remarks. The paper can be published.  

Author Response

Response to reviewer 2

The 2nd reviewer comment is:

 Authors have corrected the manuscript in accordance with all suggested remarks. The paper can be published.  

 English language and style are fine/minor spell check required

 Further to English revision within 1 revised form, the suggestion has been followed and reflected within current version.
